

# Juniper tree-ring data from the Kuramenian Mountains (Republic of Tajikistan), reveals changing summer drought signals in western Central Asia

Feng Chen[1], Tongwen Zhang[1], Andrea Seim[2], Shulong Yu[1], Ruibo Zhang[1], Hans W. Linderholm[2], Zainalobudin V. Kobuliev[3], Ahsan Ahmadov[3], Anvar Kodirov[3]

[1] Key Laboratory of Tree-ring Physical and Chemical Research of China Meteorological Administration/Xinjiang Laboratory of Tree-Ring Ecology, Institute of Desert Meteorology, China Meteorological Administration, Urumqi 830002, China,

[2] Regional Climate Group, Department of Earth Science, University of Gothenburg, Gothenburg, Sweden

[3] Institute ofWater Problems, Hydroenergy and Ecology, Academy of Science of the Republic of Tajikistan, Dushanbe 734063, Tajikistan

*Correspondence to*: Feng Chen (feng653@163.com)

**Abstract.** Coniferous forests cover the mountains in many parts of central Asia and provide large potentials for dendroclimatic studies of past climate variability. However, to date, only a few tree-ring based climate reconstructions exist from this region. Here we present a regional tree-ring chronology from moisture-sensitive *Juniperus seravschanica* from the Kuramenian Mountains (Republic of Tajikistan), which is used to reveal past summer drought variability in western Central Asia. The chronology accounts for 40.5% of the variance of the June–July self-calibrating Palmer Drought Severity Index (scPDSI) during the instrumental period (1901 to 2012). Seven dry periods including 1659–1696, 1705–1722, 1731–1741, 1758–1790, 1800–1842, 1860–1875 and



1931–1987, and five wet periods of 1742–1752, 1843–1859, 1876–1913, 1921–1930 and
1988–2015 were identified. Good agreements between drought records from western and eastern
Central Asia suggest that the PDSI records retain common drought signals and captures the
regional dry/wet periods of Central Asia. Moreover, the wavelet analysis indicates the existence of
centennial (100-150 years), decadal (50-60, 24.4 and 11.4 years) and interannual (8.0 and 2.0-3.5
years) cycles, which may linked with climate forcings, such as solar activity and ENSO. The
analysis between the scPDSI reconstruction and large-scale atmospheric circulations during the
reconstructed extreme dry and wet years can provide information about the linkages of extremes
in our scPDSI record with the Asian summer monsoon activity.
**Keywords:** Kuramenian Mountains; Tree rings; Drought reconstruction; Synoptic climatology
analysis; Tajikistan; Juniper

## 1 Introduction

As a result of climate warming during recent decades, the intensity and frequency of drought
events have been increasing (Easterling et al., 2000; Dai et al., 2011; Schrier et al., 2013). Climate
models predict a significant increase in the extent of dry areas across the globe, mainly in the
Northern Hemisphere, with an potential expansion of arid lands by up to 80% in developing
countries (Huang et al., 2015). Climate change and related drought events have significant
influences on the socioeconomic and human well-being in arid Central Asia, particularly in
densely populated dry lands, such as the Fergana Basin (Ososkova et al., 2000; Siegfried et al.,
2012; Yao et al., 2015). Lake shrinkage, oasis salinization, and water resource deterioration,
mainly due to excess water use for irrigation, have been linked to climate change, especially in the
Aral Sea Basin (Micklin, 1988; Lioubimtseva and Cole, 2006; Kezer and Matsuyama, 2006; Reyer



et al., 2015). Meteorological stations were installed at some big cities of Central Asia, such as
Samarkand, in the late 19th century, but most of observational records from the mountains areas of
Central Asia started in the 1950-1960s. Due to poor spatiotemporal coverage of meteorological
records in the mountains areas, there are uncertainties in the estimation of Central Asian climate
change. Therefore, to achieve more accurate assessments of climate change in a long-term
perspective in this region, high-resolution climate proxy data is needed.

Due to their exact dating and annual resolution, climate-sensitive trees play an important role

in providing information about past climate variability and change in many regions of the world
(Jones et al., 2009). Indeed, many of the existing long-term climate records from Central Asia
have been based on tree-ring data (Esper et al., 2001, 2002, 2003; Yuan et al., 2003; Chen et al.,
2010, 2014; Zhang et al., 2013; Solomina et al., 2014). These dendroclimatic reconstructions
allow us to better understand the spatiotemporal variations of Central Asian climate. However, the
impact of climate on tree growth can be complex, where for tree-ring formation can be influenced
by both precipitation and temperature (Fritts, 1976; Tian et al., 2007), making it difficult to
separate the precipitation signals from temperature. However, by considering monthly climate
factors and the soil moisture supply, different comprehensive drought indices, such as the
standardized precipitation evapotranspiration index (Vicente-Serrano et al., 2010) and the PDSI
(Palmer, 1965) and, have been developed. Such indices can thus be used as targets for drought
reconstructions from trees with as mixed temperature and precipitation sensitivity. . Based on large
tree-ring networks, spatial drought reconstructions have been developed for many regions,
including Europe, North America, northwestern Africa and Mongolia (e.g. Cook et al., 1999, 2010,
2015; Davi et al., 2010; Fang et al., 2010; Seftigen et al. 2015; Touchan et al., 2011). Although



some dendroclimatic studies have investigated drought variability, as well as its effect on tree
growth in Central Asia (Esper et al., 2001; Yuan et al., 2003; Chen et al., 2013, 2015a, 2015b,
2016; Seim et al., 2015, 2016), the number of tree-ring data from western Central Asia is still not
sufficient to provide a regionally comprehensive picture. . To achieve this additional
moisture-sensitive tree-ring chronologies are needed.

The Kuramenian Mountains offer good potentials for dendroclimatic study in Northern

Tajikistan. This mountain range is a source of streamflow into the small mountainous rivers in the
border areas between Tajikistan and Uzbekistan. The exploding population and scarce water
resources have stressed water supplies increasingly in the Fergana basin and its surrounding areas.
Dendroclimatic information from the Kuramenian Mountains can be used to make water resource
plans and help tackle regional climate change. This study presents a June–July PDSI
reconstruction from tree-ring width data of Turkestan juniper, obtained from two sites in the
Kuramenian Mountains, northern Tajikistan. Wavelet analysis were applied to examine any cycles
in the drought reconstruction. Furthermore, we investigated relationships between this drought
record and the Asian summer monsoon and atmospheric circulation patterns over the Pacific and
Indian Oceans.
**2 Material and methods**
2.1 Geographical settings and chronology development

The research region is located in the Kuramenian Mountains (northern Tajikistan) near the

Fergana Basin (Fig. 1), where the climate is mainly affected by the Westerlies (Chen et al., 2016).
The average annual total rainfall from the closest meteorlogical station (Khujand station, 40.22 °N,
69.73 °E, 414 m a.s.l.) amounts to 164.1 mm, with only 19.1% of the total annual rainfall falling



during the warm season which is approximately from May to September. July (average monthly
temperature of 28.6 ℃) and January (14.3 ℃) are the warmest and the coldest month, respectively
(Fig. 2). At the sampling sites (Obiasht and Adrasman), with sparse vegetation among different
trees, open-canopy juniper forests grow on thin soil (Fig. 3). All tree-ring samples were collected
from the dominant species, Zeravshan juniper (*Juniperus seravschanica*), and in total, 81 samples
(from 40 trees) were taken from the two sites. The oldest tree (1594–2015) was found at the
Adrasman site.
After drying and mounted on the mounts, tree-ring samples were polished with the 400 grit
sandpapers to enhance tree-ring boundaries. The Velmex measuring system, with a precision of
0.001 mm , was used to measured annual ring widths. The quality of the cross-dating and
measurements was controlled using the COFECHA software (Grissino-Mayer, 2001). The result
of correlation analysis reveals that high correlation ($r = 0.52$, $p < 0.001$) exists between the site
chronologies. This allowed us to use all tree-ring width series of juniper trees to construct a
regional chronology. The ARSTAN program (Cook and Kairiukstis, 1990) was used to develop a
regional chronology for the Kuramenian Mountains. Each raw ring-width series was first
detrended to remove non-climatic trends using the negative exponential curve. The standard (STD)
chronology was used in the further analyses. The fully replicated chronology with the expressed
population signal (Wigley et al., 1984) greater than or equal to 0.85 was achieved with a minimum
tree number of five trees from AD 1650.
2.2 Statistical analysis
The regional chronology was correlated with a set of monthly climate variables (including
monthly total rainfall and average temperature) from July of the previous year to September of



current year from the Khujand station for the period 1927–1990. Due to surrounding areas have
over a century of climate data, self-calibrating Palmer Drought Severity Index (scPDSI, Van der
Schrier et al., 2011) for the Kuramenian Mountains (averaged over 40.5–41.5 °N, 70.0–71.0 °E) for
the    period    1901–2012    (obtained    from    the    KNMI    Climate    Explorer    website
(http://climexp.knmi.nl/) was also used in the correlation analysis.

Correlations between the regional chronology with the monthly climate records allowed

identification of the main limiting factors for tree growth. Based on linear regression analysis, a
statistical model between the predictand (scPDSI) and the predictors (the regional chronology)
was calculated for the calibration period (1901–2012) to indicate past drought variations. A
split-sample calibration-verification test (Cook and Kairiukstis, 1990) was used to evaluate the
reliability of the scPDSI reconstruction model. The period 1901–2012 was divided into calibration
(1957–2012) and verification (1901–1956) sections. The testing statistics were employed to
evaluate model ability, including sign test (ST), coefficient of efficiency (CE) and reduction of
error (RE) (Cook and Kairiukstis, 1990). Furthermore, to investigate common drought signals
among the existing moisture-sensitive tree-ring chronologies from Western Central Asia (this
study; Seim et al., 2015; Chen et al., 2016), principal component analyses (Jolliffe, 2002) was
used over the common period (1700–2012) of tree-ring chronologies from western Central Asia (.
In this study, wet and dry periods were determined if the 31-year low-pass values were lower than
the average value of the scPDSI reconstruction continuously for more than 10 years. We also
calculated the spatial correlation using the KNMI Climate Explorer (http://climexp.knmi.nl/) to
reveal the geographical representation of our records and also investigate correlation fields with
sea surface temperature (Rayner et al., 2003). Wavelet analysis was employed to reveal any





periodicities in the scPDSI reconstruction and the temporal stability of these (Torrence and Compo,
1998). For better visual comparison, the regional drought series of western and eastern Central
Asia were standardized and smoothed with a 20-year low-pass filter. In order to explore the
linkages between reconstructed scPDSI extreme events and atmospheric circulation patterns over
West and Central Asia, NCEP climate data (Kalnay et al., 1996) were used to create May-July
composite anomaly maps of the geopotential height, SSTs and 500-hPa vector wind in the driest
10 years and wettest 10 years during the period 1948–2010.
**3. Results**
3.1 The scPDSI reconstruction

Statistical results from the ARSTAN program indicated that over the common period

1901-2015, the Kuramenian Mountains chronology had a high standard deviation (0.45),
signal-to-noise ratio (32.22) and EPS (0.97). The Variance in first the eigenvector of all series
accounted for 51.6% of the total variance, indicating that juniper tree growth at the two sites was
influenced by similar factors. Significant positive correlations ($p<0.05$)between the Kuramenian
Mountains chronology and monthly total precipitation were found in current April-July ($r$:
0.26–0.36) (Fig. 4). Significant negative correlations with monthly mean temperature were found
in current May-June ($r$: -0.28–-0.44). The Kuramenian Mountains chronology was positively and
significantly correlated with scPDSI during previous July-September, particularly from April to
September ($r$: 0.59–0.637). We also investigated the correlations between the Kuramenian
Mountainschronology and seasonally averaged scPDSI, and the strongest correlation ($r$: 0.637)
was found with mean June-July scPDSI (1901–2012). The precipitation in June to September
accounts for 7.7% of the total annual precipitation, while June-July is the hottest months. The rise





in summer (June-July) temperatures promotes evaporation, and promotes the already existing
drought stress. Thus, the water availability in summer is the main limiting factor for the juniper
tree growth. Similar moisture influences on juniper growth have also been found in high Asia
(Zhang et al., 2015; Gou et al., 2015). Thus, the scPDSI reconstruction was developed by
calibrating the Kuramenian Mountains chronology with mean June-July scPDSI data.

During the calibration period 1901–2012, the predictor variable (the Kuramenian Mountains

chronology) accounts for 40.5% of the variance in the instrumental scPDSI data (40.0% after
adjustment for loss of degrees of freedom). The positive RE and CE reveal good predictive skill of
the statistical model (Table 2). The results of the sign and first-order sign tests both exceed the
99% confidence level. These test results indicated that our statistical equation was reliable. Figure
5 shows a comparison of reconstructed and instrumental mean June-July scPDSI data in the
Kuramenian Mountains during the period 1901–2012. The comparison shows that the
reconstructed scPDSI is quite consistent with the instrumental scPDSI on short and long
timescales during the 20th century.
3.2 Analyses of drought variation in the Kuramenian Mountains

The Kuramenian Mountains reconstruction provides insight into past drought variation for

this part of northern Tajikistan during the past four centuries (Fig. 6). Dry periods occurred in CE
1659–1696, 1705–1722, 1731–1741, 1758–1790, 1800–1842, 1860–1875 and 1931–1987.
Sustained dry decades were centered on 1830 as well as around 1960. Wet periods were identified
in CE 1742–1752, 1843–1859, 1876–1913, 1921–1930 and 1988–2015. Although the period
1988–2015 was characterized by wet summers, the reconstruction shows a downward trend during
the past 10 years, which is in agreement with the observations.



The three tree-ring width chronologies of juniper trees (this study; Seim et al., 2015; Chen et
al., 2016) were correlated significantly ($p < 0.001$) among each other. The principal component
analyses indicated that the first principal component (PC1) of the three chronologies exceed an
eigenvalue of >1.5 and account for 52.53% of the total variance. Spatial climate correlation
analyses revealed that the actual (Fig. 7a) and reconstructed (Fig. 7b) scPDSI series correlate
significantly with June–July gridded scPDSI and reveal similar spatial correlation fields, albeit the
signal strength of the latter is lower. Significant positive correlations were observed in the Fergana
Basin. The significant positive correlations of PC1 and June–July gridded scPDSI are also seen
from the Fergana Basin and the neighboring areas (Fig. 7c), suggesting similar large-scale drought
influence on Western Central Asia.
During the period 1901–2015, significant positive correlations ($p < 0.05$) for the
reconstructed scPDSI series of the Kuramenian Mountains with gridded SSTs over the tropical
oceans were found after removed the linear trends of SST data (Fig. 7d). Wavelet analysis
indicated that some centennial (100-150 years), decadal (50-60, 24.3 and 11.4 year) and
interannual (8.0, 2.0-3.5 years) periodicities were found in the reconstructed scPDSI data for the
Kuramenian Mountains (Fig. 8).

## 4. Discussion


4.1 Comparing reconstructed drought in western and eastern Central Asia
Based on two moisture sensitive tree-ring chronologies from central and western Tien Shan,
China (Chen et al., 2013; Chen et al., 2015b), Chen et al. (2015b) developed a regional scPDSI
reconstruction, accounting for 70.4% of the total variance in the observations, representing eastern
Central Asia. A comparison between the Kuramenian Mountains and the eastern Central Asia



reconstructions yielded a correlation coefficient of ($r > 0.35$, $p < 0.001$, n=306). The PC1 mirrors
similar dry/wet intervals as the drought series of eastern Central Asia (Fig. 8). Common dry
periods (1710s, 1770–1780s, 1800s, 1910–1940s and 1970–1980s) and wet periods (1720–1730s,
1790s, 1850s, 1890s, 1950–1960s and 1990–2000s) in western and eastern Central Asia suggest
similar moisture variation for both regions. Some differences, existing between the drought
records (i.e. in the 1700s, 1740–1760s, 1810–1840s, 1860–1880s and 1900s), may reflect local
influences in local geography (such as the eastern Central Asia is wetter) or the difference in tree
species (juniper and spruce). Despite of this, high correlation coefficient revealed that drought
stress is the major limiting factor on the tree growth of Central Asia, and covers the whole region.
Chen et al (2015b) also found significant correlations ($p < 0.05$) between the drought series of
eastern Central Asia with gridded SSTs over the tropical ocean, very similar to what was found for
the Turkestan juniper in this study, with a strong response to SSTs. Similar patterns suggesting
that the drought variations of eastern and western Central Asia may be linked with these tropical
domains. In particular, the eastern and western Central Asia both exhibit the wetting trend during
1970–2010s, implying that a consistent moisture increase in Central Asia which is of great
significance for alleviating the serious shortage of freshwater resources.

The driest year (1917) in the Kuramenian Mountains was also found in other regions of

Central Asia (Esper et al., 2001; Chen et al., 2013, 2015b, c; Seim et al. 2015). The second driest
year (1783) of the Kuramenian Mountains coincides with the volcanic eruption of Laki (iceland)
in 1783 (Schmidt et al., 2011; Chen et al., 2012), and suggests the influence of the volcanic
eruption on the climate there. In order to further reveal the characteristics of the large-scale
extreme drought events in Central Asia, we further extracted the first principal component of the





drought series of western and eastern Central Asia which accounted for 74.8% of the total
variance during the period 1901–2005. Based on this drought series, Large-scale drought events
during the period 1916-1919, 1944-1945 and 1974-1976 were found in Central Asia. Figure 10
showed that PDSI anomalies during the period 1916-1919, 1944-1945 and 1974-1976 are
noticeable negative over central and northern Asia, and the south Asia was anomalously wet. This
suggest the presence of weak moisture transport by south Asian monsoon and the Westerlies to
central Asia, and a weak south Asian monsoon with strong moisture transport in south Asia.
4.2 Possible climate drivers
The 24.3 and 11.4-year periodicity is likely related to the variations of large-scale modes of
solar activity (Hale, 1924; Hodell et al., 2001). In eastern Central Asia, the influence of solar
cycles on drought variations has been indicated by dendroclimatic researches (e.g., Li et al., 2006).
Thus, solar activity appears to have the large-scale impacts on the drought variations of Central
Asia. Comparison of the scPDSI reconstruction and the sunspot relative number series
(http://www.sidc.be/silso/DATA/yearssn.dat) also reveals there exists a significant relationship in
the 11 year band from the 1700–2000s (Fig. 9b). Similarly, the 8.0, 3.6 and 2.1-years cycles were
linked with the variations of the cross-equatorial low level jet of the western Indian Ocean (Gong
and Luterbacher, 2008) and El Niño -Southern Oscillation (ENSO) index (Li et al., 2013) (Fig. 9c,
9d). This suggests that drought variation in Central Asia may be related to large-scale
land–atmosphere–ocean circulation systems. However, some different relationships between the
series reveal that the impacts of solar activities (i.e. in the 1900–2000s) and large-scale climate
modes on the regional drought of the Central Asia are more complicated than expected, and a
number of unknown physical processes at various timescales await further investigation.



As previously mentioned, the drought variation of Central Asia may be teleconnected with
the activity of the south Asian summer monsoon. The wet-year composite is characterized by
strengthened southerlies and westerlies entered into Central Asia associated with a negative center
over Central Asia and some positive height-anomaly centers in the Near East and Indian ocean
(Fig. 11a, b). Positive SST anomalies were found in the tropical Indian and western Pacific Ocean
during the wettest years (Fig. 11e). Relatively abundant moisture is brought across the Arabian
Peninsula and Iranian Plateau by the strong southwesterly moisture flux (Asian summer monsoon)
and traveled further northward, causing increased moisture over the southern part of central Asia.
This finding resembles previous researches that have indicated drought variations over
southwestern and central Asia are strongly linked with the West Asian subtropical westerly jet and
SSTs in the tropical Indian oceans (Mariotti, 2007; Li et al., 2010; Zhao et al., 2014).
The composite of 500-hPa geopotential height during the driest years is the reverse of the
wettest-year composite in that the negative anomaly over Central Asia is replaced by a positive
anomaly (Fig. 11d). This positive anomaly combined with a relatively low over the Near East and
Indian Ocean suggests weakened southerlies over south Asia and perhaps an enhanced dry jet
across Central Asia (Fig. 11c). Previous researches has revealed that negative SST anomalies over
the tropical Indian Ocean tend to associate with weak southwesterly winds, and lead to increased
droughts in Central Asia (Vecchi et al., 2004; Li et al., 2010). This pattern during the driest years
supports such a connection. As seen above, moisture conditions in Central Asia are linked with
SSTs in the tropical oceans and Asian summer monsoon intensity. Dendroclimatic researches
based on the improved tree-ring network should help to understand the climate mechanisms of
Central Asia.



## 5. Conclusions

In this study, based on tree-ring width series of Turkestan juniper, we developed a new June–July scPDSI reconstruction from the Kuramenian Mountains in northern Tajikistan, which indicated drought variations at different time scales over the past 366 years. The drought reconstruction captures the recent wetting trend of western Central Asia well, and represents drought variations over a large area of western Central Asia. The dry/wet periods identified in the drought reconstruction are in good agreement with drought series from eastern Central Asia. Moreover, the analysis of links between the climate variations and our scPDSI reconstruction reveals that there are some linkages of extremes in this scPDSI reconstruction with anomalous Asian summer monsoon circulation in the Indian Ocean Rim. In Central Asia, Turkestan juniper can live to about 500-1000 years (Esper et al., 2003). Thus, more efforts should be paid to extend the dendroclimatic reconstructions by collecting the cores from the old trees and develop spatial drought reconstructions to reveal the spatio-temporal drought variations of Central Asia.

**Acknowledgments**

This work was supported by NSFC Project (41405081) and National high level talents special support plan. We thank the reviewers very much whose comments greatly benefitted this manuscript.

**Contributions**

Conceived and designed the experiments: FC, ZT and KZ. Performed the experiments: FC, ZT, AA and KA. Analyzed the data: FC and ZT. Contributed reagents/ materials/analysis tools: FC,



ZT, SA, and LH. Contributed to the writing of the manuscript: FC, SA, and LH.

**Conflict of Interest Statement**
The authors declare that the research was conducted in the absence of any commercial or financial
relationships that could be construed as a potential conflict of interest.

**Data Availability**
The authors confirm that all data underlying the findings are fully available without restriction.
The PDSI reconstruction is available in the Supplement.

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













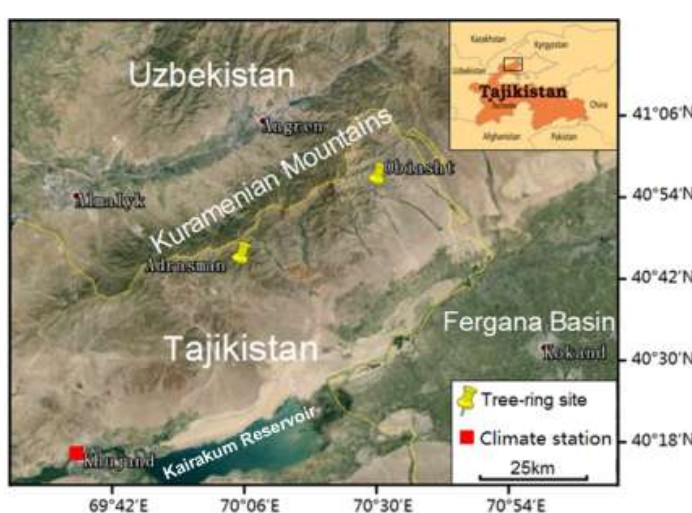


Fig. 1. Map of the climate station (Khujand) and the sampling sites in the Kuramenian Mountains,
northern Tajikistan.





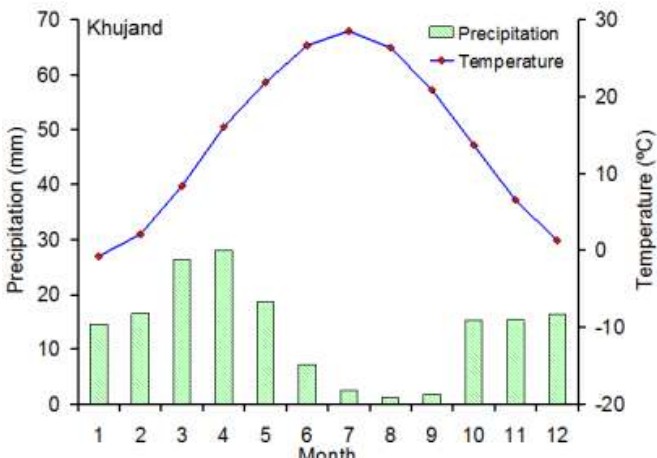


Fig. 2. Climate diagrams for the climate station of Khujand in northern Tajikistan.

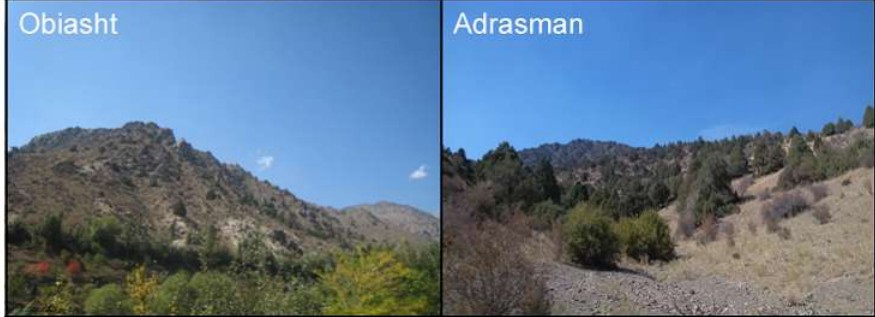


Fig. 3. Juniper trees at the different sites in the Kuramenian Mountains, northern Tajikistan.

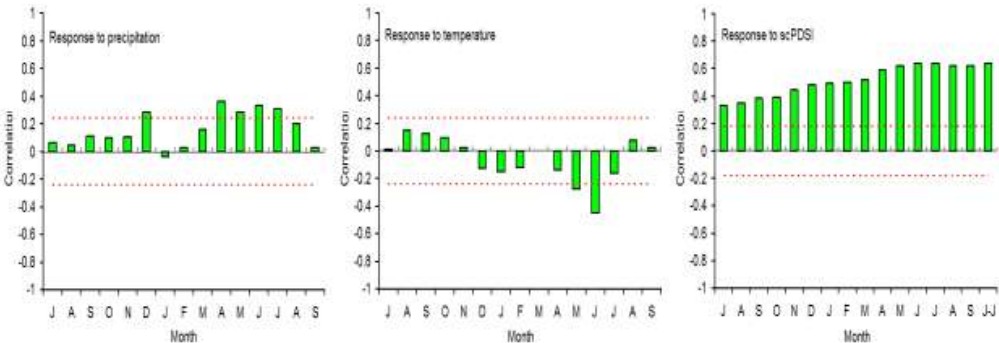



Fig. 4. Response plots for the regional chronology with monthly total precipitation (1927–1990),
mean monthly temperature (1927–1990) and monthly scPDSI (1901–2012). The coefficients were

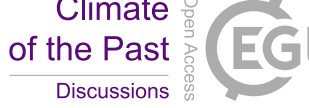


calculated from the previous July to the current September. Horizontal dashed lines denote 95%
significance levels.

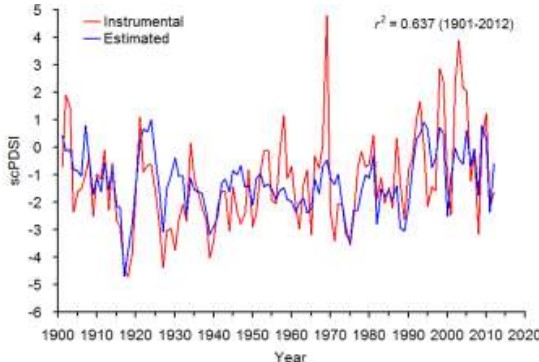


Fig. 5. Comparison between the instrumental and reconstructed mean June–July scPDSI for the
Kuramenian Mountains during the period 1901–2012.

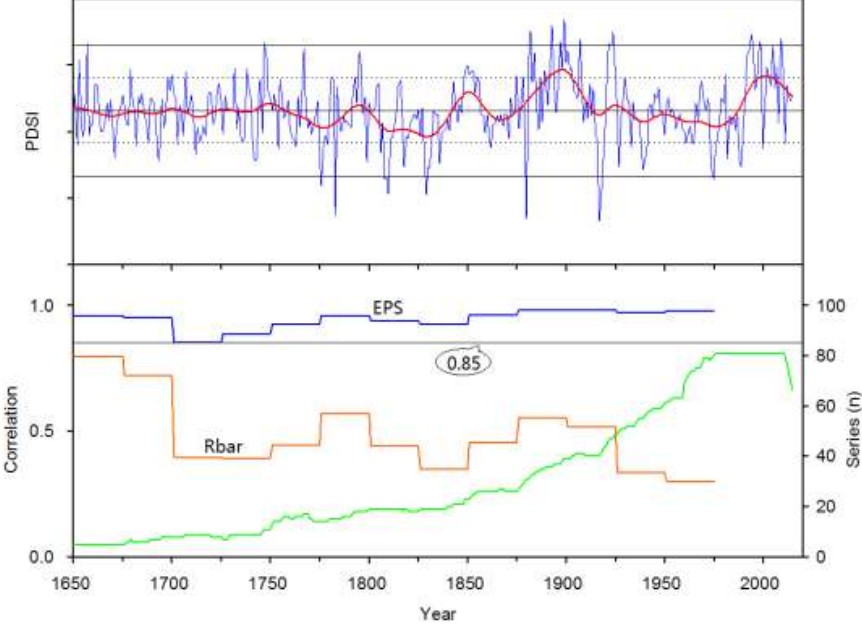


Fig. 6. Reconstructed (thin line) and 31-year low-pass filter (thick line) values of June–July
scPDSI for the Kuramenian Mountains from the regional chronology of the Kuramenian
Mountains with sample size, EPS (expressed population signal) and Rbar (average correlation
between series). Central horizontal line shows the mean of the estimated values; inner horizontal
lines (dotted lines) show the border of one standard deviation, and outer horizontal lines two
standard deviations. Rbar and EPS used moving 50-year windows, lagged 25 years.



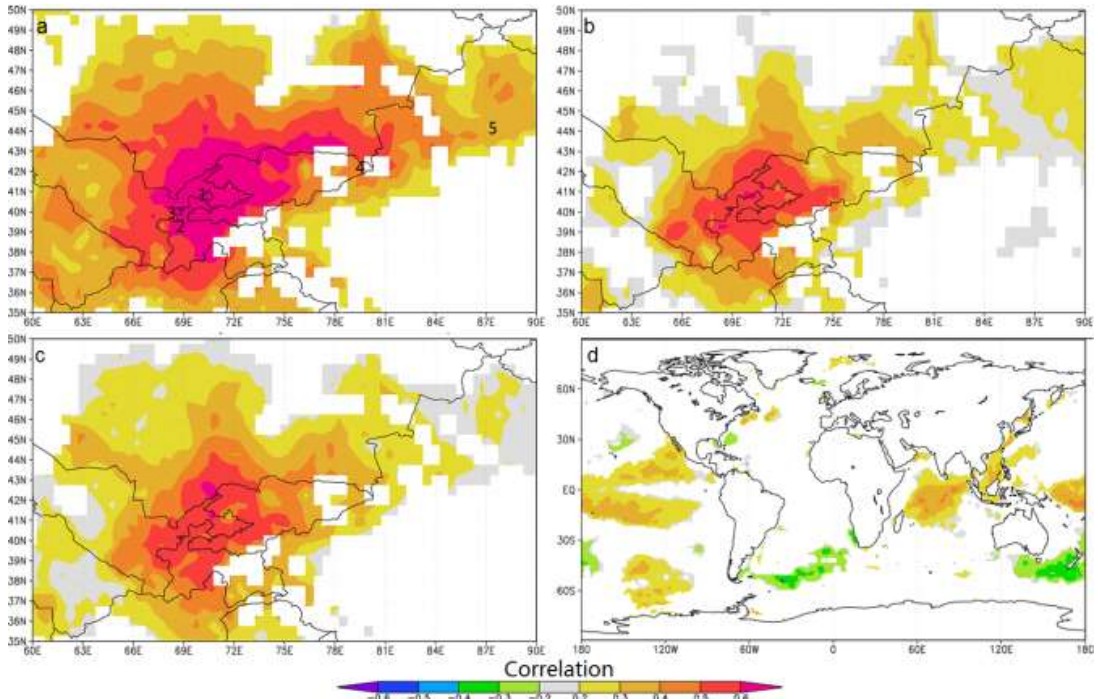

Fig. 7. Spatial correlation fields of instrumental June–July scPDSI (a), reconstructed June–July

scPDSI (b) and PC1 (c) with regional gridded June–July scPDSI for the period 1901–2012. The

numbers 1, 2, 3, 4 and 5 denote the tree ring sites of northern Tajikistan (this study), western

Tajikistan (Chen et al., 2016), Uzbekistan (Seim et al., 2015), Kyrgyzstan (Chen et al., 2013), and

China (Chen et al., 2015b). (d) Spatial Pearson correlation plots for the reconstructed June–July

scPDSI for the Kuramenian Mountains with February-July averaged HadISST1 SST after

removed the linear trends of SST data during the period 1901-2015.

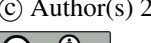

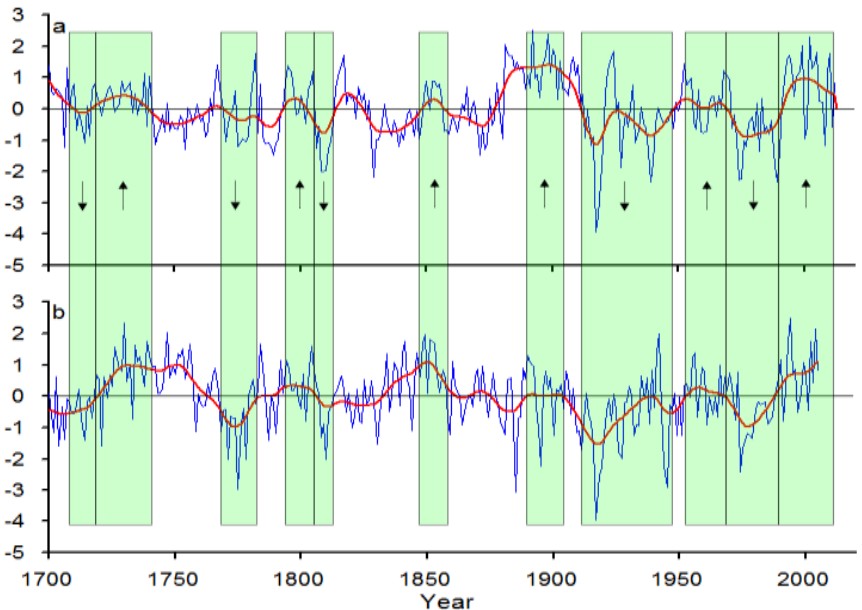


Fig. 8. Comparison of between the drought series of western (a) and eastern Central Asia (b, Chen
et al., 2015). All series were adjusted for their long-term means over the period 1700–2010, and
smoothed with a 20-year low-pass filter to emphasize long-term fluctuations. The arrows indicate
the upward/downward trends.

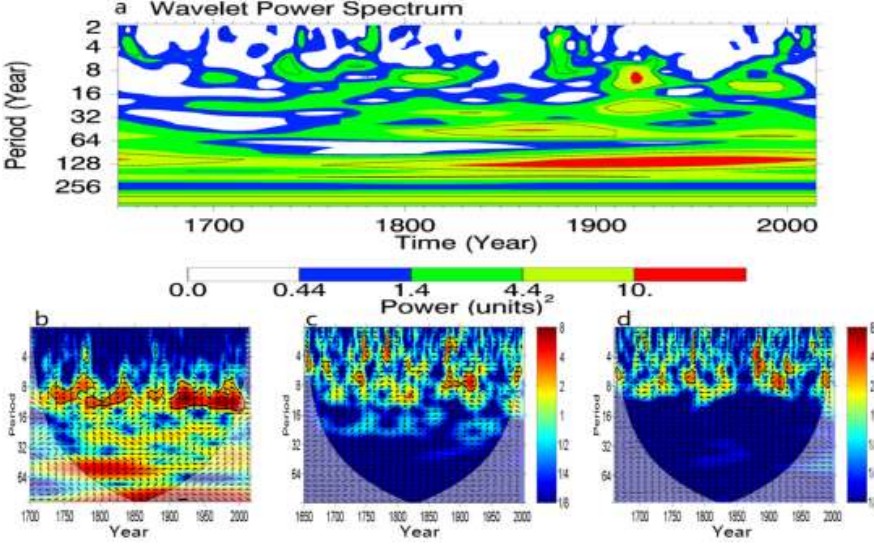


Fig. 9. (a) The wavelet power spectrum. Black contours are the 5% significant level, using a
red-noise (autoregressive lag 1) background spectrum. Cross wavelet transform of the



reconstructed    scPDSI    of    the    Kuramenian    Mountains    with    (b)    sunspot    number
(http://www.sidc.be/silso/DATA/yearssn.dat), (c) the ENSO index (Li et al., 2013) and (d) the
low-level cross-equatorial jet of the western Indian Ocean (Gong and Luterbacher, 2008). The 5%
significance level against red noise is shown as a thick contour. The relative phase relationship is
shown as arrows (with in-phase pointing to right, anti-phase pointing to left).

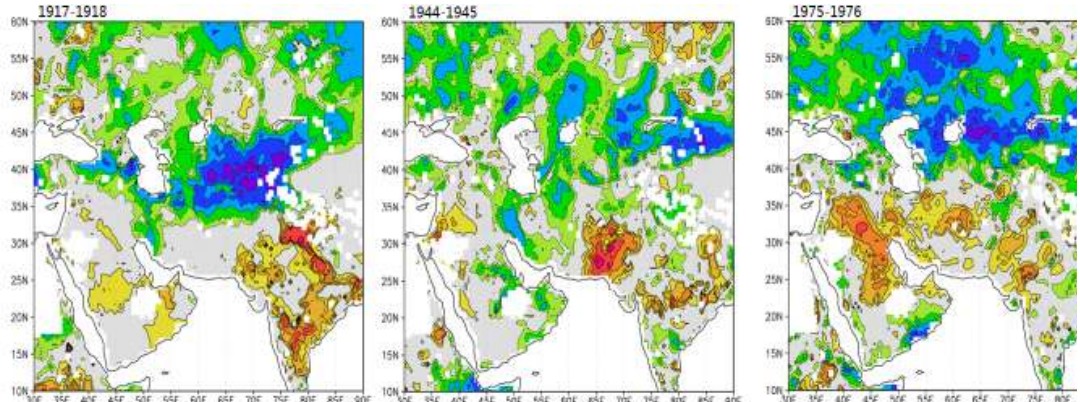


Fig. 10. PDSI anomalies during the dry period 1916-1919, 1944-1945 and 1974-1976.







Fig. 11. Composite anomaly maps of the SSTs, 500-hPa vector wind and geotpotential height

(from May to August) for the 10 wettest (a, b and e) and 10 driest (c, d and f) years for the scPDSI

reconstruction during the period 1948–2010. The five-pointed star represent the study area.


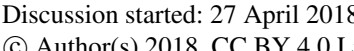


Table 1 Information about the sampling sites in the Kuramenian Mountains

| Site code | Latitude (N) | Longitude (E) | Tree number | Elevation (m) | Aspect | Slope | Species |
|-----------|--------------|---------------|-------------|---------------|--------|-------|---------|
| Obiasht | 40 ʿ52' | 70 ʿ27' | 24 | 1663.7 | E | 30 ° | *J. seravschanica* |
| Adrasman | 40 ʿ42' | 70 ʿ04' | 27 | 2035 | SE | 20 ° | *J. seravschanica* |


Table 2 Calibration and verification statistics for mean June-July scPDSI reconstructions. r:
correlation coefficient, RE: reduction of error, CE: coefficient of efficiency, ST: prediction sign
test, FST: the first-order sign test prediction sign test '+': pair of actual and predicted temperatures
showed same sign of departures from their respective mean values; '−': different sign of
departures, *Significant at the 1% level.

|  | Calibration (1957-2012) | Verification (1901-1956) | Calibration (1901-1956) | Verification (1957-2012) | Full calibration (1901-2012) |
|--|-------------------------|--------------------------|-------------------------|--------------------------|------------------------------|
| $r$ | 0.705 | 0.637 | 0.637 | 0.705 | 0.637 |
| $r^2$ | 0.410 | 0.406 | 0.406 | 0.410 | 0.406 |
| RE |  | 0.351 |  | 0.360 |  |
| CE |  | 0.282 |  | 0.329 |  |
| Sign test |  | 41+/15-* |  | 41$^+$/15$^-$* |  |
| First-order sign test |  | 45+/10-* |  | 46$^+$/9$^-$* |  |

