# Peer review of "Juniper tree-ring data from the Kuramenian Mountains (Republic of Tajikistan), reveals changing summer drought signals in western Central Asia"

_Climate of the Past, 2018_

## Referee Comment (RC1) · Anonymous Referee #1 · 8 May 2018

**Review**

*MS No.: cp-2018-44*

*"Juniper tree-ring data from the Kuramenian Mountains (Republic of Tajikistan), reveals changing summer drought signals in western Central Asia"*

*Author(s): Feng Chen, Tongwen Zhang, Andrea Seim, Shulong Yu, Ruibo Zhang, Hans W. Linderholm, Zainalobudin V. Kobuliev, Ahsan Ahmadov, and Anvar Kodirov*

The manuscript present a new 366-year series of Jun-Jul scPDSI modeled with the help of tree rings. The topic is relevant to the scope of CD.

I find that the manuscript is not strong enough for publication. It holds many unclear issues related to the reconstruction model, data analysis, and interpretations.

The paper title suggests a discussion of changing drought signals in juniper tree rings of western Central Asia, although much of the Discussion emphasizes the linkages between the Tajikistan-site PDSI reconstruction, regional PDSI pattern and atmospheric circulation.

The paper has a number of short-comings. The most obvious that the authors try to explain the variability of reconstructed moisture with ENSO, solar activity (Fig 9 cross wavelets) and volcanic eruptions (L216-217). The Discussion is lacking conclusive assertions explaining how these factors drive the moisture variability across the region.

[Figure]

The conceptual scheme linking the drought reconstruction solely to the Asian monsoon ("tropical domains") sounds speculative. How is the impact of Arctic and Atlantic air masses compatible with the Asian monsoon variability?
The tree rings collected in cold semi-arid climate is mostly influenced by the westerlies. The side map shows the position of the study area along the west-northern margin of Central Asian mountain system, where the Alay-Pamir Mountains (Tajikistan/Afghanistan) is merging with the Tian Shan Mountains (Kazakhstan/Kirgizstan).

More generally, it is unclear why the moisture fluctuations between eastern and western sub-regions of Central Asia appear so similar and coherent. It is just hard to believe that the Asian Monsoon controls the moisture regime of this entire region. The PCA analysis of the PDSI-derived moisture records must be shown and explained prior to the Discussion.

Technical flaws:

The physiological mechanism underlying the response of tree rings to moisture is not well explained and cited. There is a dozen different species of juniper trees in the studied region and their climatic response to temperature and moisture vary significantly (see Seim et al. 2016, Mukhamedshin 1980). For example, *J. seravschanica* is highly sensitive to cold but well adapted to low moisture. In opposite, *J. turkistanica* favors wet and cold habitats. *J. seravschanica* studied in the paper is strongly limited by the Apr-Sept moisture conditions (Seim et al. 2016). Why do the authors select the Jun-Jul window for their reconstruction model? How do they explain the physiological mechanism underlying the tree-ring response to soil moisture of the mid-summer months?

The reconstruction model is not clearly explained, e.g. the regression equation is not given, the residuals and quality of the model are not analyzed. Fig. 5 shows $R^2_{adj.} = 0.637$, which is actually the correlation (Table 2). The wavelet plots are unreadable due to 1) invisible arrows displaying the difference in phases (time lag) and signal coherence, and 2) missing the cone of influence (area of uncertainties). How was the periodicity of 24.3 and 11.4 yrs assessed? The Principal component analysis applied to the Tajikistan reconstructed series and Central Asian regional record (Cheng et al. 2015) is not shown in the Results. Fig. 10 is missing scale bar and location of the study. Abstract and Results have no indication for the span of reconstructed series. Notice that the sampling was done in the Kuramin Range. Calling this range "Kuramenian Mountains" is nor correct.

---

## Referee Comment (RC2) · Feng Chen et al. · 31 May 2018

General comment

The paper "Juniper tree-ring data from the Kuramenian Mountains (Republic of Tajikistan), reveals changing summer drought signals in western Central Asia" by F. Chen et al. is devoted to reconstruct past summer drought variability (PDSI based) in western Central Asia (actually, the authors analyzed a very local area in the Kuramenian Mountains which contains just two sample plots).

Overall impression of the work is very mixed. The authors use traditional techniques to

analyze their dendroclimatical datasets and to obtain a local PDSI reconstruction and its analysis. As an example based on well-known "classical' procedure they obtained tree-ring measurements from 81 juniper trees located at the elevations from 1600 to 2035 m. But what is a reason to mix them together? Early it was shown a tree-ring response on climate can be different for mountain regions and significantly depended on site elevations (e.g. Touchan et al., 2016 for vast part of Eastern Mediterranean). That response can be changed each 500 m of additional elevation. It means that the moisture on 2000 meters can be different from the values on 1000 meters. At the same time Chen F. and co-authors try to extend their results for the very diverse (in context of elevation) and vast region such as western Central Asia. Could the authors prove that the western Central Asia is a more homogeneous in comparison with the semi-arid Eastern Mediterranean in the context of the tree-ring response on climate depended on altitude (or elevation)?

Other serious issue of the manuscript is a way to use different approaches which are not appropriate to obtain the certain results (see specific comments).

Speculation concerning the global climate patterns and their connections with the obtained reconstruction should be clarified or proven taking into account wavelet features (see specific comments). For example, why the correlation between reconstructed scPDSI and sunspot number becomes much stronger in XX centuries in the high-frequency domain (Fig. 9b)? How that phenomena can be explained in terms of climatology?

I recommend to re-submit the paper after major revision.

Specific comments

Lines 70-72 Authors wrote: "To achieve this additional moisture-sensitive tree-ring chronologies are needed." What does "moisture-sensitive tree-ring chronologies" mean? Is the local tree-ring signal sensitive to soil moisture or to mixed signal "precipitation-temperature", i.e. PDSI associated? Could the authors clarify it?

Lines 103-104 Authors wrote: "Each raw ring-width series was first detrended to re-move non-climatic trends using the negative exponential curve." It was shown early (i.e. Melvin, 2004) that the selected standardization can be a reason of "divergence problem"? What was a criteria to select "the negative exponential curve" as a stan-dardization method?

Lines 109-110 Authors wrote: "The regional chronology was correlated with a set of monthly climate variables (including monthly total rainfall and average temperature) from July..." What was a criteria to mix (average) tree-ring indexes from two different plots located on different elevation levels? The elevation difference is more than 500 m. Early it was shown tree-ring response is significantly different for different eleva-tions and depended on site elevation for the extensive area of Eastern Mediterranean (Touchan et al., 2016). Can the authors prove the tree-ring signal are the same for the both sites? If they are able to show it then they can go further.

Line 126 Authors wrote: "...principal component analyses (Jolliffe, 2002)..." Could the authors include PCA statistics in the MS to understand why and how new PCA components can be associated with "common drought signals"?

Line 128 Authors wrote: "In this study, wet and dry periods were determined if the 31-year low-pass values..." Why the "31-years low-pass filter" is selected? I am sure in case of other window for filter we can obtain other wet and dry periods.

Lines 132-133 Authors wrote: "Wavelet analysis was employed to reveal any period-icities in the scPDSI reconstruction..." What was a kind of wavelet analysis used to "reveal any periodicities" taking into account that in most cases the wavelet technique allows to obtain a frequency strongly affected by the time window?

Line 135 Authors wrote: "...smoothed with a 20-year low-pass filter." Why was 20-years filter used? What will be a difference in case of 15-, 21-, 25-years filters used?

Line 144 Authors wrote: "...signal-to-noise ratio (32.22) and EPS (0.97)." What was

a value of Rbar between individual trees for both sites? It seems to me the Rbar was pretty low (about 0.3 or less).

Lines 144-145 Authors wrote: "The Variance in first the eigenvector of all series accounted for 51.6% of the total variance, . . ." What does "all series" mean? Are the time series indexed or raw? How the first PC is corresponding to regional chronology?

Lines 153-158 It seems to me the lines 153-158 is not a result and they should be removed to discussion section.

Line 164 Authors wrote: "These test results indicated that our statistical equation was reliable". Where is the statistical equation or equations? The authors used cross-validation approach to testify the model. They calibrated the model on the 1957-2012 and verified it on the 1901-1956 as a first step. Then they used an inverse approach (to change the calibration and verification periods). It means they obtained 2 equations as minimum (see table 2). How are the equations statistically different or the same? Which equation is used for reconstruction? And what does it mean "common calibration period 1901-2012"? Does it mean the third equation?

Lines 177-178 Authors wrote: "The three tree-ring width chronologies of juniper trees (this study; Seim et al., 2015; Chen et al., 2016) were correlated significantly (p < 0.001) among each other." What are the correlation values between chronologies? What is the common time period?

Lines 189-192 Authors wrote: "Wavelet analysis indicated that some centennial (100-150 years), decadal (50-60, 24.3 and 11.4 year) and interannual (8.0, 2.0-3.5 years) periodicities were found in the reconstructed scPDSI data for the Kuramenian Mountains (Fig. 8)." It seems to me the wavelet analysis is not a best choice to analyze the periodicity in time series taking into account the wavelet features in time and frequency domains. For example, multi-taper method could be more appropriate in that case.

---

## Author Comment (AC1) · 20 Jul 2018

Dear reviewer1

First of all, we thank you for your valuable advice, which will help us to further improve this article. The manuscript present a new 366-year series of Jun-Jul scPDSI modeled with the help of tree rings. The topic is relevant to the scope of CD. I find that the manuscript is not strong enough for publication. It holds many unclear issues related to the reconstruction model, data analysis, and interpretations.The paper title suggests

a discussion of changing drought signals in juniper tree rings of western Central Asia, although much of the Discussion emphasizes the linkages between the Tajikistan-site PDSI reconstruction, regional PDSI pattern and atmospheric circulation. Response: Yes, this article has some shortcomings, but it is a standardized method for reconstruction models and data analysis, and it does not have fatal defects. At the same time, the purpose of our reconstruction is not only to reveal the facts of regional climate change, but more importantly to reveal the mechanism of climate change and serve to improve climate simulation and strategies to deal with climate change.

The paper has a number of short-comings. The most obvious that the authors try to explain the variability of reconstructed moisture with ENSO, solar activity (Fig 9 cross wavelets) and volcanic eruptions (L216-217). The Discussion is lacking conclusive assertions explaining how these factors drive the moisture variability across the region. Response: Indeed, we only objectively demonstrated the relationship between them and did not conduct a mechanism analysis. In fact, a large number of studies have been conducted in the past to analyze the effects of ENSO, volcanic activity and solar activity on tree rings and climate. But as you know, there is very little research on tree wheel climate in this area, and this study only shows preliminary results. If we can get a chance to modify it, we will explain the mechanism further in the article.

The conceptual scheme linking the drought reconstruction solely to the Asian monsoon ("tropical domains") sounds speculative. How is the impact of Arctic and Atlantic air masses compatible with the Asian monsoon variability? Response: This area is affected by a variety of climate circulation, forming a climate characteristic similar to that of the Iranian plateau, and is very different from the Tianshan Mountains. Under the influence of the meridional circulation, the Southwest monsoon (moisture) crossed Southwest Asia into Central Asia. We will explain the mechanism further in the article.

The tree rings collected in cold semi-arid climate is mostly influenced by the westerlies. The side map shows the position of the study area along the west-northern margin of Central Asian mountain system, where the Alay-Pamir Mountains (Tajikistan/Afghanistan) is merging with the Tian Shan Mountains (Kazakhstan/Kirgizstan). More generally, it is unclear why the moisture fluctuations between eastern and western sub-regions of Central Asia appear so similar and coherent. It is just hard to believe that the Asian Monsoon controls the moisture regime of this entire region. The PCA analysis of the PDSI-derived moisture records must be shown and explained prior to the Discussion. Response: No, over the past eight years, we has found that some areas are relatively wet and can grow spruce (see figure), which is affected by Marine climate, and. But in tajikistan and southern kyrgyzstan, eastern uzbekistan is drier, summer rains are rare and forests grow only on the windward slopes of high mountains. As you can see, our research area is located in the south slope. Only will there be enough water vapor to meet the growth needs of trees when the southern monsoon and the westerly wind system work together. The two regions are connected, so their climate is of course consistent. The monsoon is only likely to affect the southern slope of the area, and in the north it is affected by the western wind. I can improve.

Technical flaws: The physiological mechanism underlying the response of tree rings to moisture is not well explained and cited. There is a dozen different species of juniper trees in the studied region and their climatic response to temperature and moisture vary significantly (see Seim et al. 2016, Mukhamedshin 1980). For example, J. seravschanica is highly sensitive to cold but well adapted to low moisture. In opposite, J. turkistanica favors wet and cold habitats. J. seravschanica studied in the paper is strongly limited by the Apr-Sept moisture conditions (Seim et al. 2016). Why do the authors select the Jun-Jul window for their reconstruction model? How do they explain the physiological mechanism underlying the tree-ring response to soil moisture of the mid-summer months? Response: Indeed. In different growing environments, trees have different responses to climate. In order to reconstruct drought changes, we only chose dry sampling sites. In Dr Seim's study, they collected data from a large number of sampling sites and analyzed the climate response characteristics of Juniper at different altitudes and environments. Because the months is the most important growing season for plants and crops, we chose June-July PDSI as target. The mechanism is

well understood, because this is the peak season for forest growth in high mountains, and there is very little rainfall in this area. This has been explained in this paper, and the variance of the reconstruction equation in this period is highest. I also will improved.

The reconstruction model is not clearly explained, e.g. the regression equation is not given, the residuals and quality of the model are not analyzed. Fig. 5 shows R2 adj. =0.637, which is actually the correlation (Table 2). Response: The model will be added in the paper. We use the standard reconstruction method and process, and show the results of equation test. I don't know why you would say we didn't show the test of the equation.

The wavelet plots are unreadable due to 1) invisible arrows displaying the difference in phases (time lag) and signal coherence, and 2) missing the cone of influence (area of uncertainties). How was the periodicity of 24.3 and 11.4 yrs assessed? Response: the periodicity of 24.3 and 11.4 was determined by calculating his highest peak. I have shown the meaning and scope of the arrows in the diagram. I don't know why the wavelet plots are unreadable. Could you provide an example diagram. The Principal component analysis applied to the Tajikistan reconstructed series and Central Asian regional record (Cheng et al. 2015) is not shown in the Results. Response: Yes, I will add the result section.

Fig. 10 is missing scale bar and location of the study. Response: Yes, I will improve the fig. 10.

Abstract and Results have no indication for the span of reconstructed series. Notice that the sampling was done in the Kuramin Range. Calling this range "Kuramenian Mountains" is nor correct. Response: Yes, I will improve. The name of the mountains is very confusing. According to the local map of tajikistan and some tourist information, we adopted this name. But according to the information you provided, we can modify it. https://www.advantour.com/tajikistan/nature/mountains.htm

[Figure]

**Fig. 1.**

---

## Author Comment (AC2) · 20 Jul 2018

First of all, we thank you for your valuable advice, which will help us to further improve this article. General comment The paper "Juniper tree-ring data from the Kuramenian Mountains (Republic of Tajikistan), reveals changing summer drought signals in western Central Asia" by F. Chen et al. is devoted to reconstruct past summer drought variability (PDSI based) in western Central Asia (actually, the authors analyzed a very local area in the Kuramenian Mountains which contains just two sample plots). Overall impression of the work is very mixed. The authors use traditional techniques to

analyze their dendroclimatical datasets and to obtain a local PDSI reconstruction and its analysis. As an example based on well-known "classical' procedure they obtained tree-ring measurements from 81 juniper trees located at the elevations from 1600 to 2035 m. But what is a reason to mix them together? Early it was shown a tree-ring response on climate can be different for mountain regions and significantly depended on site elevations (e.g. Touchan et al., 2016 for vast part of Eastern Mediterranean). That response can be changed each 500 m of additional elevation. It means that the moisture on 2000 meters can be different from the values on 1000 meters. At the same time Chen F. and co-authors try to extend their results for the very diverse (in context of elevation) and vast region such as western Central Asia. Could the authors prove that the western Central Asia is a more homogeneous in comparison with the semi-arid Eastern Mediterranean in the context of the tree-ring response on climate depended on altitude (or elevation)? Response: Although it is a regional study, because it is the most densely populated area in Central Asia and a key area, it is important for the sustainable development of Central Asia. Due to the fact that the precipitation in the area is very small in summer, it is basically not due to differences in altitude. Since the data correlation of these sampling points is relatively good, cross-dating has been carried out. At the same time, we can see that this reconstruction sequence represents large-scale drought signals.

Other serious issue of the manuscript is a way to use different approaches which are not appropriate to obtain the certain results (see specific comments). Speculation concerning the global climate patterns and their connections with the obtained reconstruction should be clarified or proven taking into account wavelet features (see specific comments). For example, why the correlation between reconstructed scPDSI and sunspot number becomes much stronger in XX centuries in the highfrequency domain (Fig. 9b)? How that phenomena can be explained in terms of climatology? Response: Yes, although many studies have confirmed that solar activity has a significant impact on regional climate and tree growth, few studies have analyzed the mechanisms. If I get the opportunity to modify it, I will analyze the mechanism of these relationships

I recommend to re-submit the paper after major revision. Specific comments Lines 70-72 Authors wrote: "To achieve this additional moisture-sensitive tree-ring chronologies are needed." What does "moisture-sensitive tree-ring chronologies" mean? Is the local tree-ring signal sensitive to soil moisture or to mixed signal "precipitation-temperature", i.e. PDSI? Could the authors clarify it? Response: Yes, due to the lack of precipitation in this area and weather stations in mountainous areas, the signals here are generally mixed signals, so we only reconstruct the region PDSI, instead of choosing precipitation. I will improve.

Lines 103-104 Authors wrote: "Each raw ring-width series was first detrended to remove non-climatic trends using the negative exponential curve." It was shown early (i.e. Melvin, 2004) that the selected standardization can be a reason of "divergence problem"? What was a criteria to select "the negative exponential curve" as a standardization method? Response: Yes, this is based on our sampling and tree growth curve. All of our tree-ring growth curves conform to the negative exponential function model. We have deleted abnormal sequences.

Lines 109-110 Authors wrote: "The regional chronology was correlated with a set of monthly climate variables (including monthly total rainfall and average temperature) from July. . ." What was a criteria to mix (average) tree-ring indexes from two different plots located on different elevation levels? The elevation difference is more than 500 m. Early it was shown tree-ring response is significantly different for different elevations and depended on site elevation for the extensive area of Eastern Mediterranean (Touchan et al., 2016). Can the authors prove the tree-ring signal are the same for the both sites? If they are able to show it then they can go further. Response: I can make a list of their relationships and charts to prove that there is a strong common signals between them.

Line 126 Authors wrote: ". . .principal component analyses (Jolliffe, 2002). . ." Could the authors include PCA statistics in the MS to understand why and how new PCA components can be associated with "common drought signals"? Response: Yes, I

can improve. To investigate common drought signals among the tree-ring chronologies (spruce and juniper) from Western Central Asia, I did the PCAfor large-scale.

Line 128 Authors wrote: "In this study, wet and dry periods were determined if the 31-year low-pass values. . ." Why the "31-years low-pass filter" is selected? I am sure in case of other window for filter we can obtain other wet and dry periods. Response: Yes, you are right. Due to the difference in window, there will be a difference in dry and wet periods. But we needed to get more low-frequency signals, so we tried multiple Windows, we compared the results, we chose the 31-year window.

Lines 132-133 Authors wrote: "Wavelet analysis was employed to reveal any period-icities in the scPDSI reconstruction. . ." What was a kind of wavelet analysis used to "reveal any periodicities" taking into account that in most cases the wavelet technique allows to obtain a frequency strongly affected by the time window? Response: Yes, I used morlet wavelet, I will improve.

Line 135 Authors wrote: ". . .smoothed with a 20-year low-pass filter." Why was 20-years filter used? What will be a difference in case of 15-, 21-, 25-years filters used? Response: We compared the results, we chose the 20-year window.

Line 144 Authors wrote: ". . .signal-to-noise ratio (32.22) and EPS (0.97)." What was a value of Rbar between individual trees for both sites? It seems to me the Rbar was pretty low (about 0.3 or less). Response: The correlations were both found to be significant at the 0.001 level.

Lines 144-145 Authors wrote: "The Variance in first the eigenvector of all series ac-counted for 51.6% of the total variance, . . ." What does "all series" mean? Are the time series indexed or raw? How the first PC is corresponding to regional chronology? Response: Yes , all time series indexed for the regional series. The correlation is 0.96 between the first PC and regional chronology.

Lines 153-158 It seems to me the lines 153-158 is not a result and they should be

removed to discussion section. Response: I will improve.

Line 164 Authors wrote: "These test results indicated that our statistical equation was reliable". Where is the statistical equation or equations? The authors used crossvalidation approach to testify the model. They calibrated the model on the 1957-2012 and verified it on the 1901-1956 as a first step. Then they used an inverse approach (to change the calibration and verification periods). It means they obtained 2 equations as minimum (see table 2). How are the equations statistically different or the same? Which equation is used for reconstruction? And what does it mean "common calibration period 1901-2012"? Does it mean the third equation? Response: Yes, you and the first reviewer both pointed out this shortcoming. If you get a chance to modify it, I 'll show you the reconstruction equation and the test results. No, I just used one model for 1901-2012

Lines 177-178 Authors wrote: "The three tree-ring width chronologies of juniper trees (this study; Seim et al., 2015; Chen et al., 2016) were correlated significantly (p < 0.001) among each other." What are the correlation values between chronologies? What is the common time period? Response: Yes , it is easy. I will add. The correlation is over 0.4, the common period is 1700-2013.

Lines 189-192 Authors wrote: "Wavelet analysis indicated that some centennial (100-150 years), decadal (50-60, 24.3 and 11.4 year) and interannual (8.0, 2.0-3.5 years) periodicities were found in the reconstructed scPDSI data for the Kuramenian Mountains (Fig. 8)." It seems to me the wavelet analysis is not a best choice to analyze the periodicity in time series taking into account the wavelet features in time and frequency domains. For example, multi-taper method could be more appropriate in that case. Response: Yes, I can do multi-taper analysis with wavelet analysis.

---

## Author Comment (AC3) · 28 Aug 2018

Response letter

The manuscript present a new 366-year series of Jun-Jul scPDSI modeled with the help of tree rings. The topic is relevant to the scope of CD.

I find that the manuscript is not strong enough for publication. It holds many unclear issues related to the reconstruction model, data analysis, and interpretations.The paper title suggests a discussion of changing drought signals in juniper tree rings of western Central Asia, although much of the Discussion emphasizes the linkages between the Tajikistan-site PDSI reconstruction, regional PDSI pattern and atmospheric circulation.

**Response:** First of all, we thank you for your valuable advice, which will help us to further improve this article. Yes, this article has some shortcomings, but it is a standardized method for reconstruction models and data analysis, and it does not have fatal defects. At the same time, the purpose of our reconstruction is not only to reveal the facts of regional climate change, but more importantly to reveal the mechanism of climate change and serve to improve climate simulation and strategies to deal with climate change.

The paper has a number of short-comings. The most obvious that the authors try to explain the variability of reconstructed moisture with ENSO, solar activity (Fig 9 cross wavelets) and volcanic eruptions (L216-217). The Discussion is lacking conclusive assertions explaining how these factors drive the moisture variability across the region.

**Response:** Indeed, we only objectively demonstrated the relationship between them and did not conduct a mechanism analysis. In fact, a large number of studies have been conducted in the past to analyze the effects of ENSO, volcanic activity and solar activity on tree rings and climate. But as you know, there is very little research on tree-ring climate in this area, and this study only shows preliminary results. If we can get a chance to modify it, we will explain the mechanism further in the discussion, and added some important references.

The conceptual scheme linking the drought reconstruction solely to the Asian monsoon ("tropical domains") sounds speculative. How is the impact of Arctic and Atlantic air masses compatible with the Asian monsoon variability?

**Response:** This area is affected by a variety of climate circulation, forming a climate characteristic similar to that of the Iranian plateau, and is very different from the Tianshan Mountains. Under the influence of the meridional circulation, the Southwest monsoon (moisture) crossed Southwest Asia into Central Asia. We will explain this mechanism further in the discussion.

The tree rings collected in cold semi-arid climate is mostly influenced by the westerlies. The side map shows the position of the study area along the west-northern margin of Central Asian mountain system, where the Alay-Pamir Mountains (Tajikistan/Afghanistan) is merging with the Tian Shan Mountains (Kazakhstan/Kirgizstan). More generally, it is unclear why the moisture fluctuations between eastern and western sub-regions of Central Asia appear so similar and coherent. It is just hard to believe that the Asian Monsoon controls the moisture regime of this entire region. The PCA analysis of the PDSI-derived moisture records must be shown and explained prior to the Discussion.

**Response:** No, over the past eight years, we has found that some areas are relatively wet and can grow spruce (see figure), which is affected by Marine climate, and. But in Tajikistan and southern Kyrgyzstan, eastern Uzbekistan is drier, summer rains are rare and forests grow only on the windward slopes of high mountains. As you can see, our research area is located in the south slope. Only will there be enough water vapor to meet the growth needs of trees when the southern monsoon and the westerly wind system work together. The two regions are connected, so their climate is of course consistent. The monsoon is only likely to affect the southern slope of the area, and in the north it is affected by the western wind. I will discuss the impact of the interaction between the West wind and the monsoon on the climate in Central Asia in the mechanism exploration section. At the same time, I will cite the papers published in recent years on the precipitation mechanism in Central Asia. The results of these papers can support our conclusions.

Technical flaws:

The physiological mechanism underlying the response of tree rings to moisture is not well explained and cited. There is a dozen different species of juniper trees in the studied region and their climatic response to temperature and moisture vary significantly (see Seim et al. 2016, Mukhamedshin 1980). For example, J. seravschanica is highly sensitive to cold but well adapted to low moisture. In opposite, J. turkistanica favors wet and cold habitats. J. seravschanica studied in the paper is strongly limited by the Apr-Sept moisture conditions (Seim et al. 2016). Why do the authors select the Jun-Jul window for their reconstruction model? How do they explain the physiological mechanism underlying the tree-ring response to soil moisture of the mid-summer months?

**Response:** Indeed. In different growing environments, trees have different responses to climate. In order to reconstruct drought changes, we only chose dry sampling sites. In Dr Seim's study, they collected data from a large number of sampling sites and analyzed the climate response characteristics of Juniper at different altitudes and environments. Because the months are the most important growing season for plants and crops, we chose June-July PDSI as target. The mechanism is well understood, because this is the peak season for forest growth in high mountains, and there is very little rainfall in this area. This has been explained in this paper, and the variance of the reconstruction equation in this period is highest. I will improved the response of tree rings, and discuss about the link of tree rings with to soil moisture.

The reconstruction model is not clearly explained, e.g. the regression equation is not given, the residuals and quality of the model are not analyzed. Fig. 5 shows R2 adj. =0.637, which is actually the correlation (Table 2).

**Response:** Yes, you are right. The model will be added in the paper. We use the standard reconstruction method and process, and show the results of equation test. I don't know why you would say we didn't show the test of the equation. But we decided to modify the content about the quality test of the equation.

 The wavelet plots are unreadable due to 1) invisible arrows displaying the difference in phases (time lag) and signal coherence, and 2) missing the cone of influence (area of uncertainties). How was the periodicity of 24.3 and 11.4 yrs assessed?

**Response:** the periodicity of 24.3 and 11.4 was determined by calculating his highest peak. I have

shown the meaning and scope of the arrows in the diagram. I don't know why the wavelet plots are unreadable. Could you provide an example diagram? Or I will use multi-taper methods to analysis cycles based on the advice of reviewer 2.

The Principal component analysis applied to the Tajikistan reconstructed series and Central Asian regional record (Cheng et al. 2015) is not shown in the Results.

**Response:** Yes, I will show the results of the principal component analysis applied to the Tajikistan reconstructed series and Central Asian regional records in the result section.

Fig. 10 is missing scale bar and location of the study.

**Response:** Yes, I will redraw the fig. 10 and added scale bar and location of the study.

Abstract and Results have no indication for the span of reconstructed series. Notice that the sampling was done in the Kuramin Range. Calling this range "Kuramenian Mountains" is nor correct.

**Response:** Yes, I will improve. The name of the mountains is very confusing. According to the local map of tajikistan and some tourist information, we adopted this name. But according to the information you provided, we will modify the name in the fig 1.

https://www.advantour.com/tajikistan/nature/mountains.htm

---

## Author Comment (AC4) · 28 Aug 2018

Response letter

**Response:** First of all, we thank you for your valuable advices, which will help us to further improve this article.

General comment

The paper "Juniper tree-ring data from the Kuramenian Mountains (Republic of Tajikistan), reveals changing summer drought signals in western Central Asia" by F. Chen et al. is devoted to reconstruct past summer drought variability (PDSI based) in western Central Asia (actually, the authors analyzed a very local area in the Kuramenian Mountains which contains just two sample plots). Overall impression of the work is very mixed. The authors use traditional techniques to analyze their dendroclimatical datasets and to obtain a local PDSI reconstruction and its analysis. As an example based on well-known "classical' procedure they obtained tree-ring measurements from 81 juniper trees located at the elevations from 1600 to 2035 m. But what is a reason to mix them together? Early it was shown a tree-ring response on climate can be different for mountain regions and significantly depended on site elevations (e.g. Touchan et al., 2016 for vast part of Eastern Mediterranean). That response can be changed each 500 m of additional elevation. It means that the moisture on 2000 meters can be different from the values on 1000 meters. At the same time Chen F. and co-authors try to extend their results for the very diverse (in context of elevation) and vast region such as western Central Asia. Could the authors prove that the western Central Asia is a more homogeneous in comparison with the semi-arid Eastern Mediterranean in the context of the tree-ring response on climate depended on altitude (or elevation)?

**Response:** Although it is a regional study, because it is the most densely populated area (**Fergana Basin**) in Central Asia and a key area, it is important for the sustainable development of Central Asia. Yes, this paper is not just a traditional tree-ring study. We hope to use some method to make our article more persuasive. We 'll straighten out the text and make it smoother

As you pointed out, this is a regional study. Due to the distance between these sites are relatively short, and the environment is similar and the same tree species, we consider the establishment of a regional chronology to obtain regional drought signals. Meanwhile, since the correlations between these individual chronologies are high, the cross-dating data from individual sites were combined into regional chronology. I 'll describe the chronology development process in detail in the methods and results section.

About climate homogeneities. Yes, differences in topography and altitude can lead to large climatic differences in different regions, which can affect the response of tree growth. Because this area is an inland arid area, the climate difference is smaller than that of the coastal area. We will try to use some weather station data at different altitudes and tree-ring series to prove that the western Central Asia is a more homogeneous in comparison with the semi-arid Eastern Mediterranean in the context of the tree-ring response on climate depended on altitude.

Other serious issue of the manuscript is a way to use different approaches which are not appropriate to obtain the certain results (see specific comments). Speculation concerning the global climate patterns and their connections with the obtained reconstruction should be clarified or proven taking into account wavelet features (see specific comments). For example, why the correlation between reconstructed scPDSI and sunspot number becomes much stronger in XX

centuries in the highfrequency domain (Fig. 9b)? How that phenomena can be explained in terms of climatology?

**Response:** I admit that there is not enough evidence in our article to explain the mechanism of climate change now. If I were given the opportunity to revise it, I would intensify my analysis of the climate mechanism, especially the focus of the article: the links between the South Asian monsoon region and the changes in dry and wet in Central Asia and its mechanism. We will also look for further evidence of the effects of solar activity on the local climate to support our conclusions and refine our statements and discussions. I agree with you, and the interpretation of the climate mechanism will be the focus of our revision

I recommend to re-submit the paper after major revision.

Specific comments

Lines 70-72 Authors wrote: "To achieve this additional moisture-sensitive tree-ring chronologies are needed." What does "moisture-sensitive tree-ring chronologies" mean? Is the local tree-ring signal sensitive to soil moisture or to mixed signal "precipitation-temperature", i.e. PDSI? Could the authors clarify it?

**Response:** Yes, due to the lack of precipitation in this area and weather stations in mountainous areas, the signals here are generally mixed signals, so we only reconstruct the region PDSI, instead of choosing precipitation. In the result section, I will make a detailed revision to the analysis of the tree-ring's climate response and highlighted the advantages of the comprehensive indicators.

Lines 103-104 Authors wrote: "Each raw ring-width series was first detrended to remove non-climatic trends using the negative exponential curve." It was shown early (i.e. Melvin, 2004) that the selected standardization can be a reason of "divergence problem"? What was a criteria to select "the negative exponential curve" as a standardization method?

**Response:** Yes, this is based on our sampling and tree growth curve. All of our tree-ring growth curves conform to the negative exponential function model. You are an experienced tree-ring expert. The use of de-trend methods has a great influence on the results of dendroclimatic studies. We also tried a variety of de-trending methods and chose this one.I will describe the method and the process of establishing the chronology in detail in the method and the result section.

Lines 109-110 Authors wrote: "The regional chronology was correlated with a set of monthly climate variables (including monthly total rainfall and average temperature) from July. . ." What was a criteria to mix (average) tree-ring indexes from two different plots located on different elevation levels? The elevation difference is more than 500 m. Early it was shown tree-ring response is significantly different for different elevations and depended on site elevation for the extensive area of Eastern Mediterranean (Touchan et al., 2016). Can the authors prove the tree-ring signal are the same for the both sites? If they are able to show it then they can go further.

**Response:** Yes, climate consistency is very important for the spatial representation of single-point climate reconstruction. We will try to use some weather station data at different altitudes and tree-ring series to prove that the western Central Asia is a more homogeneous in comparison with the semi-arid Eastern Mediterranean in the context of the tree-ring response on climate depended on altitude. At the same time, add the correlation analysis of single point

chronology in the result part.

Line 126 Authors wrote: ". . .principal component analyses (Jolliffe, 2002). . ." Could the authors include PCA statistics in the MS to understand why and how new PCA components can be associated with "common drought signals"?

**Response:** Yes, to investigate common drought signals among the tree-ring chronologies (spruce and juniper) from Western Central Asia, I did the PCA for large-scale. I will introduce the principal component analysis in the method section and discuss the representation of common signals in the result section

Line 128 Authors wrote: "In this study, wet and dry periods were determined if the 31-year low-pass values. . ." Why the "31-years low-pass filter" is selected? I am sure in case of other window for filter we can obtain other wet and dry periods.

**Response:** Yes, you are right. Due to the difference in window, there will be a difference in dry and wet periods. But we needed to get more low-frequency signals, so we tried multiple Windows, we compared the results, we chose the 31-year window. Or we can analyze extreme values, or we can use moving averages.

Lines 132-133 Authors wrote: "Wavelet analysis was employed to reveal any periodicities in the scPDSI reconstruction. . ." What was a kind of wavelet analysis used to "reveal any periodicities" taking into account that in most cases the wavelet technique allows to obtain a frequency strongly affected by the time window?

**Response:** Yes, I used morlet wavelet, and will describe the wavelet analysis method in detail in the method section. And revise the discussion section to enhance the discussion of the impact of the cycle on regional climate change

Line 135 Authors wrote: ". . .smoothed with a 20-year low-pass filter." Why was 20-years filter used? What will be a difference in case of 15-, 21-, 25-years filters used?

**Response:** We compared the results, we chose the 20-year window. The effect on the final result is not very large, and the retention of regional low-frequency signals is better. I will make improvements to the window setting based on the light of other experts

Line 144 Authors wrote: ". . .signal-to-noise ratio (32.22) and EPS (0.97)." What was a value of Rbar between individual trees for both sites? It seems to me the Rbar was pretty low (about 0.3 or less).

**Response:** The correlations were both found to be significant at the 0.001 level. I 'll show you the quality of the regional chronology in the outcome section.

Lines 144-145 Authors wrote: "The Variance in first the eigenvector of all series accounted for 51.6% of the total variance, . . ." What does "all series" mean? Are the time series indexed or raw? How the first PC is corresponding to regional chronology?

**Response:** Yes, all tree-ring series indexed for the regional series. The correlation is 0.96 between the first PC and regional chronology. I will add the first principal component to the results in response to the tree ring chronology and indicate the meaning of all sequences

Lines 153-158 It seems to me the lines 153-158 is not a result and they should be removed to discussion section.

**Response:** I will improve and move the content to discussion section.

Line 164 Authors wrote: "These test results indicated that our statistical equation was reliable". Where is the statistical equation or equations? The authors used crossvalidation approach to testify the model. They calibrated the model on the 1957-2012 and verified it on the 1901-1956 as a first step. Then they used an inverse approach (to change the calibration and verification periods). It means they obtained 2 equations as minimum (see table 2). How are the equations statistically different or the same? Which equation is used for reconstruction? And what does it mean "common calibration period 1901-2012"? Does it mean the third equation?

**Response:** Yes, you and the first reviewer both pointed out this shortcoming. A split calibration-verification scheme was employed to test the reliability of the reconstruction. The period from 1957 to 2012 was used for calibration and 1901–1956 for verification; this process was then reversed. Precipitation data for the full 1901–2012 period was then used to calibrate the final reconstruction If I get a chance to modify it, I 'll show you the reconstruction equation and the test results in result section.

Lines 177-178 Authors wrote: "The three tree-ring width chronologies of juniper trees (this study; Seim et al., 2015; Chen et al., 2016) were correlated significantly (p < 0.001) among each other." What are the correlation values between chronologies? What is the common time period?

**Response:** Yes, this is my omission. I will add the results of the correlation analysis to my article and add relevant data. The correlation is over 0.4, the common period is 1700-2013.

Lines 189-192 Authors wrote: "Wavelet analysis indicated that some centennial (100-150 years), decadal (50-60, 24.3 and 11.4 year) and interannual (8.0, 2.0-3.5 years) periodicities were found in the reconstructed scPDSI data for the Kuramenian Mountains (Fig. 8)." It seems to me the wavelet analysis is not a best choice to analyze the periodicity in time series taking into account the wavelet features in time and frequency domains. For example, multi-taper method could be more appropriate in that case.

**Response:** Yes, Your opinion is correct. Since the periodic signals are not very strong, the results shown in Figure 8 are not ideal. We will use multi-taper analysis in the article, while retaining the contents of cross wavel transform.